# Influence of Electric Wing Tip Propulsion on the Sizing of the Vertical Stabilizer and Rudder in Preliminary Aircraft Design

Alexander Albrecht [1,*], Andreas Bender [1], Philipp Strathoff [2], Clemens Zumegen [2], Eike Stumpf [2] and Andreas Strohmayer [1]

1    Institute of Aircraft Design, University of Stuttgart, 70569 Stuttgart, Germany
2    Institute of Aerospace Systems, RWTH Aachen University, 52062 Aachen, Germany
*    Correspondence: albrecht@ifb.uni-stuttgart.de; Tel.: +49-711-685-62409

**Abstract:** During preliminary aircraft design, the vertical tail sizing is conventionally conducted by the use of volume coefficients. These represent a statistical approach using existing configurations' correlating parameters, such as wing span and lever arm, to size the empennage. For a more detailed analysis with regard to control performance, the vertical tail size strongly depends on the critical loss of thrust assessment. This consideration increases in complexity for the design of the aircraft using wing tip propulsion systems. Within this study, a volume coefficient-based vertical tail plane sizing is compared to handbook methods and the possibility to reduce the necessary vertical stabilizer size is assessed with regard to the position of the engine integration and their interconnection. Two configurations, with different engine positions, of a hybrid-electric 19-seater aircraft, derived from the specifications of a Beechcraft 1900D, are compared. For both configurations two wiring options are assessed with regard to their impact on aircraft level for a partial loss of thrust. The preliminary aircraft design tool MICADO is used to size the four aircraft and propulsion system configurations using fin volume coefficients. These results are subsequently amended by handbook methods to resize the vertical stabilizer and update the configurations. The results in terms of, e.g., operating empty mass and mission fuel consumption, are compared to the original configurations without the optimized vertical stabilizer. The findings support the initial idea that the connection of the electric engines on the wing tips to their respective power source has a significant effect on the resulting torque around the yaw axis and the behaviour of the aircraft in case of a power train failure, as well as on the empty mass and trip fuel. For only one out of the four different aircraft designs and wiring configurations investigated it was possible to decrease the fin size, resulting in a 53.7% smaller vertical tail and a reduction in trip fuel of 4.9%, compared to the MICADO design results for the original fin volume coefficient.

**Keywords:** aircraft design; distributed propulsion; hybrid-electric flight; critical loss of thrust; vertical stabilizer




## 1. Introduction

Aircraft design is a complex and interrelated field of research. Every modification of a specific component in the design process has an influence on other components or the aircraft/platform itself. This leads to an iterative design process, ideally resulting in one converged optimum design for the given top level aircraft requirements (TLARs). Small, evolutionary technology changes mostly result in designs close to already known configurations, emphasizing today's well-chosen concepts. This changes for revolutionary technology steps, such as the implementation of distributed (electric) propulsion (DEP). Several designs, differing radically from today's configurations, have already been investigated in order to use the full technological potential enabled through the use of DEP on aircraft. Globally, there have been many research attempts trying to make use of DEP's aerodynamic potential. The NASA X-57 design uses the velocity increase downstream

of a propeller to reduce the wing surface [1]. Wing tip propellers are also used by other configurations in order to reduce the wake-induced drag, such as the NASA Pegasus [1], the Airbus/TBM/Safran EcoPulse [2] and the European research project FUTPRINT50 [3]. All of these projects feature designs with unconventional propeller positions for their propulsion system integration. This raises the question of the suitability of a volume coefficient-based vertical tailplane (VTP) sizing which is based on statistics of existing conventional aircraft without DEP. This is also valid for the configurations investigated in this paper.

The study presented is part of the GNOSIS project, which aims for a holistic evaluation of the potential of propulsion system electrification for commercial passenger aircraft with seat capacities in the range of 9 to 50 seats. In the first project phase, the focus is on the conceptual design of a 19-seater aircraft. The following study therefore compares a volume coefficient-based sizing of the VTP during preliminary aircraft design with handbook methods. This is performed for a partial turboelectric 19-seater incorporating wing tip propulsion with regard to the impact of the inboard engine position and the interconnection of the propulsors and thereafter the impact of a reduction in the required VTP size on the aircraft.

According to [4], the VTP is usually sized by two major flight conditions. One being the operation with one engine inoperative and the other with maximum cross-wind capability. The focus of the study presented here is the investigation of the usability of the given VTP in case of engine failure-induced thrust asymmetry and the possible reduction in VTP size and its impact on the preliminary aircraft design. As investigated by Hoogreef and Soikkeli [5], as well as Vechtel and Buch [6] and proven by Schneider et al. [7] in a full-scale flight test, the directional stability may also be provided by the use of differential thrust. Therefore, the influence of the directional stability on the vertical tail size is neglected in this study.

For conventional configurations, the dimensioning contributor for a critical loss of thrust (CLT), or "one engine inoperative" (OEI) scenario has been the control authority of the VTP around the yaw axis. Particularly for DEP configurations, the limiting CLT condition might also be given by the maximum aileron authority in certain configurations. For the study presented in this paper, only the VTP's yaw control authority shall be assessed, as the considered positions of the propulsion system integration lead to an aircraft, for which the resulting yaw moments are significantly larger than the roll moments [6].

The contents of the paper are structured as follows. First the preliminary aircraft design tool used and the methodology for calculation of the aircraft and vertical tail parameters are explained. Then, a short overview of the considered aircraft configurations is given. Thereafter, the results of the optimization of the vertical stabilizer size for the different aircraft are given. It is shown how the fin size changes depending on the propeller positions and electrical interconnections. Moreover, certain architectures lead to uncontrollable aircraft designs in case of a critical loss of thrust, refraining from the additional implementation of components and hence additional weight. The calculated VTP size is then used to determine the impact on the aircraft using a mission data analysis of the individual configurations, hence accounting for the interrelated effects of a possible reduction in VTP size on the preliminary aircraft design.

*A Short Discussion of "Critical Loss of Thrust"*

With the change in the EASA CS23 from Amendment 4 [8] to Amendment 5 [9], the definition of the engine failure has changed. The former and well defined "failure of the critical engine" has been replaced by "critical loss of thrust" (CLT). On purpose, this term has not been specified any further. Therefore, the need to find a solution in order to assess distributed propulsion configurations and to define a "critical loss of thrust" scenario in accordance with EASA CS23.2115 [9] arises. An evaluation conducted by NASA with regard to the respective U.S. American specifications (which are almost identical to the European specifications), NASA experts have identified several certification gaps concerning both

the propulsion system and the whole aircraft certification in their analysis [10]. In order to overcome the propulsion system-related gaps, they recommend to conduct a Markov analysis to identify the most likely powertrain failure. Especially for hybrid configurations, Markov analysis can lead to the result that all powertrain components contribute to a system failure probability of less than $10 \times 10^{-9}$ [11]. Therefore, the results of the Markov analysis suggest to neglect a deeper investigation of further CLT scenarios. A different approach to define the most critical scenario is found in Jézégou et al. [12]. First, top level aircraft functions (TLAFs) are identified. Next, the impact of a failure is correlated with these TLAFs in order to assess the criticality of the failure. For a design incorporating different energy generation paths, additionally the investigation should include multiple and possible cascading failures in order to identify the most critical scenario and the influence on the TLAFs. A TLAF interaction investigated in this paper is the suitability of the vertical tailplane (VTP) to counteract the resulting CLT torque, having been sized using conventional handbook methods within a preliminary aircraft design process. Based on the wiring possibilities depicted in Figure 1, a failure in the combustion engine also results in a failure of the connected electrical propulsor. Therefore, the impact of the CLT scenario on the required VTP size has to be taken into account within the aircraft design process.

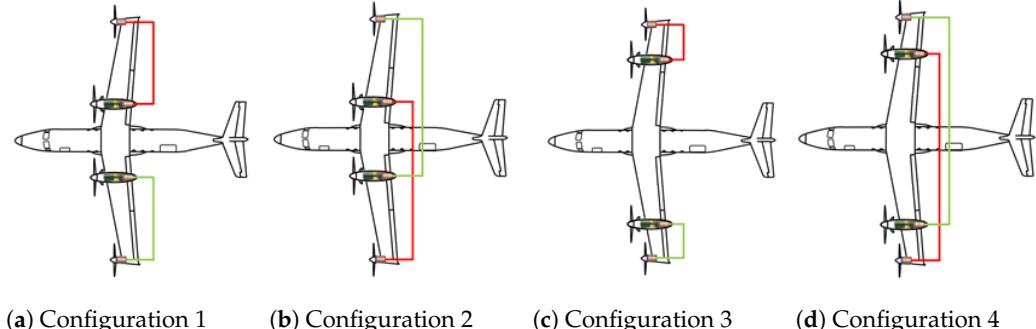

|  (**a**) Configuration 1  |  (**b**) Configuration 2  |  (**c**) Configuration 3  |  (**d**) Configuration 4  |

**Figure 1.** PT2025 aircraft with on-wing (Configuration 1) and cross wiring option (Configuration 2), as well as the PT2025opt aircraft with on-wing (Configuration 3) and cross wiring option (Configuration 4), with the red and green lines symbolizing the independent wiring harnesses.

This is especially interesting for the provided GNOSIS configurations, as, due to their powertrain layout, the probability of a 50% thrust loss is equal to the probability for the loss of the combustion engine only, assumed to be around $\times 10^{-5}$. Therefore, in contrast to [6], the loss of two propulsors has to be investigated with regard to the torque they impose on the VTP. The most critical case for a CLT is considered to be the "go" case, where the engine failure occurs at a speed above V1 and the takeoff run has to be continued. Therefore, according to [6] the maximum thrust at $V_{mc} = 1.2 \cdot V_s$ is used within the scope of this paper.

## 2. Methodology

In this section, a short overview of the analysed aircraft and wiring options, as well as the used software tools and the theoretical backgrounds used in the later evaluation, is given. The conceptional aircraft design tasks are performed using the aircraft design and evaluation environment MICADO, developed by the Institute of Aerospace Systems of the RWTH Aachen University, to iterate the complete aircraft design, while the rudder sizing is performed via MATLAB®-based tools.

### 2.1. Overview of the Conceptual Aircraft Design Tool MICADO

For the analysis of the aircraft, the multidisciplinary-integrated conceptual aircraft design and optimization (MICADO) environment is used [13,14]. MICADO is an extended version of the university conceptual aircraft design and optimization (UNICADO) environment [15] and was developed at the RWTH Aachen University's Institute of Aerospace Systems (ILR) in 2008.

As shown in Figure 2, the MICADO aircraft design process features an iterative part including aircraft component sizing, detailed system design and a design analysis step. The iteration is repeated until selected parameters of the aircraft, e.g., maximum takeoff mass (MTOM), operating empty mass (OME), mission fuel mass and lateral position of the aircraft's centre of gravity, do not change more than user-defined margins between two consecutive iteration steps.

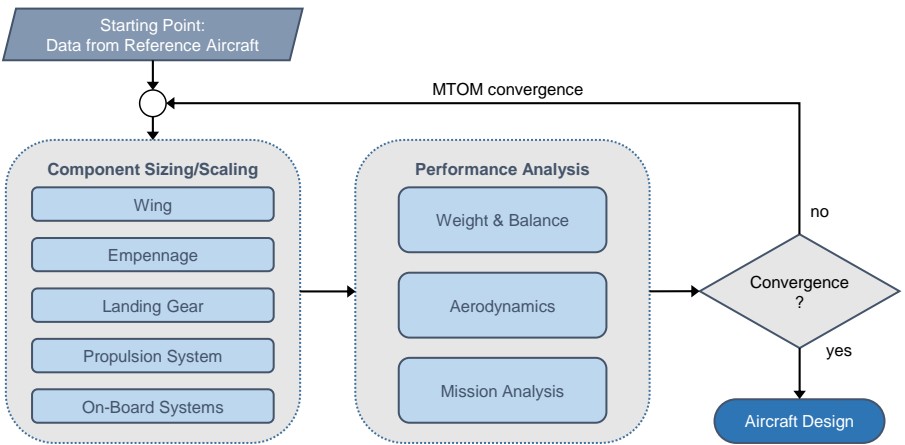

**Figure 2.** MICADO processing flow overview.

As comprehensive descriptions of MICADO and its tools can be found in the previously mentioned references, the following paragraphs focus on the MICADO tools and equations most important for this study.

The volume coefficient of the vertical tail determines its area. The value of the volume coefficient is either set by the user or calculated using an existing tail geometry.

Whereas mass estimation of most aircraft components, such as fuselage, landing gear, tail plane, is performed using semi-empirical handbook methods, wing mass estimation stands out as a tool based on analytical and semi-empirical methods. This tool takes into account the effect of point masses representing propulsion system components on the resulting wing structure and mass [16]. The secondary wing structure mass and mass penalties were estimated with semi-empirical methods; however, mass penalties due to aeroelastic effects were not included. Since the tail plane and electrical conductors are the focus of this study, applied methodologies for the estimation of their masses are presented in the following. The approach for estimation of the mass of the tail surfaces is taken from [17]. The fin mass is calculated in pounds according to Equation (1) and later converted to kilograms,

$$W_{fin} = 2.62 \cdot S_V + 1.5 \cdot 10^{-5} \cdot \frac{N_{ult} \cdot b_V^3 \cdot (8.0 + 0.44 \cdot \frac{MTOM}{S_{ref}})}{(t/c)_{avg} \cdot cos^2 \Lambda_{ea}} \tag{1}$$

where $S_V$ is the fin area including the rudder, $N_{ult}$ the ultimate load factor, $b_V$ the span of the fin, $S_{ref}$ the reference wing area, $(t/c)_{avg}$ the average airfoil thickness of the fin and $\Lambda_{ea}$ the average sweep of the quarter chord line.

For calculation of the power cable masses according to Stückl [18], the length of the conductor using the positions of the components connected by each conductor, and the power to be transferred (in terms of current $I$ and voltage $U$) by each conductor is taken into account. Moreover, for calculation of the overall conductor mass, the copper wire itself as well as insulation and sheath materials are considered. Assumptions regarding material constants and equations can be found in Table 1.

**Table 1.** Assumptions and equations for conductor design.

| Component | Density (kg/m³) | Thickness/ Diameter (mm) |
|---|---|---|
| Wire | 8920 | $D_{wire} = \sqrt{\frac{4 \cdot 0.0144 \cdot I[A]^{1.4642}}{\pi}}$ |
| Insulation | 930 | $t_{insulation} = 0.2325 \cdot U[kV] + 1.73682$ |
| Sheath | 930 | $t_{sheath} = 0.035 \cdot D_{wire}[mm] + 1$ |

Within the aerodynamic performance estimation module, lift and induced drag of the clean wing configuration are calculated using the German Aerospace Centre's LIFTING_LINE [19]. This program is able to consider the propeller-induced velocities on wing aerodynamics. Propeller-induced velocities are calculated for cruise conditions using a blade element momentum theory. The results are added to the LIFTING_LINE inputs. A more detailed description of this process can be found in [16]. Remaining drag components of the lifting surfaces and the remaining aircraft components are estimated using semi-empirical handbook methods. Most important for the study at hand is the estimation of the fin's viscous drag, calculated following an approach by Raymer [20]:

$$C_{D,Fin} = \frac{C_f \cdot FF \cdot Q \cdot S_{wet,Fin}}{S_{ref}} \tag{2}$$

where $C_F$ is the flat-pate skin-friction drag coefficient of the fin, $FF$ is the form factor of the fin, $Q$ is a interference factor, $S_{wet}$ is the fin's wetted area and $S_{ref}$ is the wing's reference area.

The entire design and convergence process with the modules employed for this study is shown in Figure 2 above.

*2.2. Reference and Concept Aircraft*

Based on the outcome of a market analysis, the Beechcraft 1900D was chosen as the reference aircraft. According to the results of an initial technology identification and selection process (with respect to technologies associated with aircraft propulsion system electrification), a partial turboelectric propulsion system featuring two gas turbines supplemented with two electrically driven propellers on the wing tips for a reduction in induced drag was selected as the most promising concept for the evaluation in a year-2025-scenario [21]. The basic assumption in the technology selection process was that an aircraft with electrified propulsion systems needs to fulfil identical mission requirements as current conventional aircraft in terms of its range, speed and passenger capacity in order to be recognized as a viable alternative.

Based on data of the Beechcraft 1900D and the PT6A-67D turboprop engines available to the public, such as the Pilot Operating Handbook [22] and the EASA-issued engine type certificate [23], a redesign of this aircraft was conducted using the MICADO environment [15], see Figure 3 left. In order to obtain a comparable conventional reference aircraft, some adjustments were made prior to the electrification of the Beechcraft 1900D. First, the wing was moved from a low to a high position to ensure sufficient ground clearance for the wing tip propellers. Corresponding to this, the position of the landing gear was moved from the wing to the fuselage. Lastly, the engine performance and the scaling factors of the mass estimation methodologies were adjusted according to the description in Section 2.1. The resulting aircraft from a further execution of the aircraft design loop after these changes served as the conventional reference for subsequent comparisons, depicted in the centre-left of Figure 3.

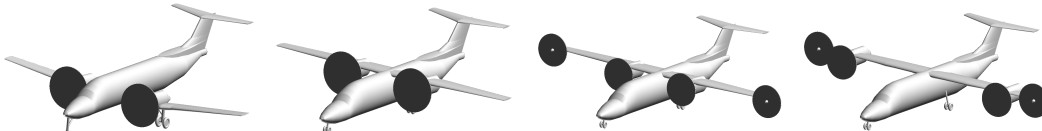

**Figure 3.** Redesign of a Beechcraft 1900D (**left**), modified redesign (**centre-left**), as well as non-optimized PT2025 (**centre-right**) and optimized PT2025opt (**right**) versions of the partial turboelectric concept aircraft.

In a second major modification step, the conventional propulsion system was exchanged with the partial turboelectric propulsion system, leading to the PT2025 aircraft configuration, see Figure 3 centre-right. The positions of the gas turbines of this aircraft do not change compared to the conventional reference aircraft. They are supplemented by two electric wing tip propellers that are powered directly from two generators mechanically connected to the gas turbines. Subsequently, an aerodynamic optimization of the propeller positions revealed a further outboard location of the conventionally driven propellers, leading to the derivative aircraft design PT2025opt [16]. Since MICADO considers only static loads, the estimation of the wing mass was conducted based on the sizing of the wing box structure considering the three quasi-static load cases, pull-up, gust and landing. Aeroelastic investigations were conducted by numerical flutter analysis and showed that no critical flutter instabilities occur up to 1.2 times the dive speed. Figure 3 shows the optimized aircraft configuration on the right-hand side.

For all aircraft configurations the same design mission, 510 NM trip with 100 NM diversion distance reserve and a 45 min holding, was used. Cruise altitude was set to 23,000 ft at a cruise speed of Mach 0.4.

*2.3. Considered Wiring Configurations*

The partial turboelectric powertrain architecture in this study uses four propellers, two driven by electric motors and two driven by gas turbines. Each electric motor is directly coupled via cables to its related generator, which converts part of the shaft power of the gas turbine, sparing the use of large electric energy storage devices. This configuration enables two wiring options, as suggested by [21]. An "on-wing wiring" approach, where the gas turbine on one wing is coupled to the electric motor on the same wing and a "cross wiring (x-wiring)" approach where the gas turbine is coupled to the electric motor on the opposite wing. Both wiring options can be seen in Figure 1 for the PT2025 and PT2025opt configuration, respectively. The red and green lines symbolize two separate and independent wiring harnesses, each connecting one gas turbine driven generator to one electric motor.

Some parameters of the different configurations are listed in Table 2. All four aircraft are similar to each other and have roughly the same wingspan, differing only 0.1 m. The PT2025opt aircraft are lighter than the PT2025 aircraft with the same wiring option, as the further outboard positioning of the gas turbine engines reduces the structural mass of the wing. The different positions of the gas turbine can also be seen in the conductor mass, where the difference between the "on-wing" and "cross-wing" option is larger for the PT2025opt aircraft due to the increased difference in the necessary cable lengths that can be seen in Figure 1. The increased aerodynamic efficiency of the PT2025opt aircraft, as mentioned in Section 2.2, can be seen in the trip fuel mass. The PT2025opt aircraft with cross wiring (Configuration 4) has a higher OME and MTOM than the PT2025 aircraft with on-wing wiring (Configuration 1), but uses 1.0% less fuel over the same mission.

**Table 2.** Parameters of the investigated concept aircraft.

| | PT2025 | | PT2025opt | |
| | On-Wing (Configuration 1) | Cross-Wing (Configuration 2) | On-Wing (Configuration 3) | Cross-Wing (Configuration 4) |
| --- | --- | --- | --- | --- |
| Wingspan | 17.3 m | 17.3 m | 17.2 m | 17.3 m |
| OME | 4892 kg | 4931 kg | 4825 kg | 4927 kg |
| MTOM | 7629 kg | 7671 kg | 7537 kg | 7653 kg |
| Conductor mass | 31.4 kg | 56.6 kg | 14.2 kg | 71.8 kg |
| Trip fuel | 606 kg | 609 kg | 591 kg | 600 kg |

### 2.4. Handbook Methods for Vertical Tail Plane Sizing According to Roskam

For the resizing of the vertical stabilizer, methods from Part II [24] and Part VI [25] of Roskam's book series on airplane design are used. This series covers the whole design process of an aircraft, from the preliminary design phase to the detailed construction of the different components and serves as standard literature in aircraft design. Most of the formulas from Roskam used in this paper are empirical correlations mainly derived from the DATCOM study of the United States Air Force [19], where an extensive number of experiments to gather data and to derive empirical equations was performed. As the methods used in this study are intended to be used within a preliminary aircraft design and iterated intensively during the mission data interpolation, the computation performance requirements should be kept at a minimum. According to Ciliberti et al. [4] this can be achieved using handbook methods. The error contained within the DATCOM method and hence their derivatives, Roskam [24,25] being one of them, lays around $-3.0\%$ for vertical tail aspect ratios of 1.0. As the investigated aspect ratios are around 0.7, the error can be neglected for the configuration in this paper.

According to Roskam [24], a first idea of the required VTP size for conventional configurations can be estimated using the so called volume coefficient, defined as

$$\bar{V}_v = x_v \cdot S_v / S_{ref} \cdot b \tag{3}$$

where $\bar{V}_v$ is the vertical tail volume coefficient, $S_v$ the VTP area, $S_{ref}$ the reference wing area, $b$ the wing span and $x_v$ the longitudinal distance from the aircraft's centre of gravity (CG) to the VTP's aerodynamic centre. As seen in Equation (3) the formula lacks any significance concerning the controllability of the investigated design and is solely derived from the evaluation of mostly conventional configurations, already in service.

In Part II "Preliminary configuration design and integration of the propulsion system" [24], the rudder deflection required to keep the aeroplane stable in case of OEI is given as

$$\delta_r = (N_D + N_{t_{crit}}) / (q_{mc} \cdot S_{ref} \cdot b \cdot C_{n_{\delta_r}}) \tag{4}$$

with the reference wing area $S_{ref}$, the wing span $b$ and the yawing moment of the remaining engine(s) $N_{t_{crit}}$. According to [24], the yawing moment resulting from the parasitic drag increase in the inoperative engine $N_D$ for a propeller-driven aircraft with variable pitch propellers is 0.1 times $N_{t_{crit}}$. The dynamic pressure $q_{mc}$ is calculated at the minimum control speed $v_{mc} = 1.2 \cdot v_s$, with $v_s$ being the lowest stall speed. $C_{n_{\delta_r}}$ is the so-called control power derivative. The value of $\delta_r$ should not exceed $25°$ [24].

The control power derivative can be obtained via the following formula from Roskam's Part VI [25]:

$$C_{n_{\delta_r}} = -C_{y_{\delta_r}} \cdot (l_v \cdot cos\alpha + z_v \cdot sin\alpha) / b \tag{5}$$

where $l_v$ and $z_v$ are the horizontal and vertical distances, respectively, between the CG of the aircraft and the aerodynamic centre ($AC_v$) of the vertical tail and $\alpha$ is the angle of attack of the aircraft. The "side-force-due-to-rudder" derivative $C_{y_{\delta_r}}$ is calculated via:

$$C_{y_{\delta_r}} = C_{L_{\alpha_v}} \cdot k' \cdot K_b \cdot ((\alpha_\delta)_{C_L} / (\alpha_\delta)_{c_l}) \cdot (\alpha_\delta)_{c_l} \cdot S_v / S_{ref}. \tag{6}$$

where $S_v$ is the surface area of the vertical tail and the coefficients $k'$, $K_b$, $((\alpha_\delta)_{C_L}/(\alpha_\delta)_{c_l})$, $(\alpha_\delta)_{c_l}$ and the lift curve slope of the vertical tail $C_{L_{\alpha v}}$ can be obtained from Roskam [pp. 228–261] [25].

$$C_{L_{\alpha v}} = 2\pi \cdot A_{v_{eff}}/(2 + \sqrt{A_{v_{eff}}^2 \cdot \beta^2/k^2 \cdot (1 + tan^2(\Lambda_{c/2})/\beta^2) + 4}) \tag{7}$$

with the semi-chord sweep angle of the vertical tail $\Lambda_{c/2}$ and

$$\beta = \sqrt{1 - M_{mc}^2} \tag{8}$$

$$k = c_{l_{\alpha M}}/2\pi \tag{9}$$

where $M_{mc}$ is the Mach number corresponding to the minimum control speed. $c_{l_{\alpha M}}$ is the lift curve slope of the VTP at the same Mach number which can be calculated from

$$c_{l_{\alpha M}} = c_{l_\alpha}/\sqrt{1 - M_{mc}^2}. \tag{10}$$

The effective aspect ratio of the vertical tail $A_{v_{eff}}$ in Equation (7) is obtained via

$$A_{v_{eff}} = (A_{v(f)}/A_v) \cdot A_v \cdot (1 + K_{vh} \cdot (A_{v(hf)}/A_{v(f)} - 1)) \tag{11}$$

with the vertical tail aspect ratio

$$A_v = b_v^2/S_v \tag{12}$$

where $b_v$ and $S_v$ are the span and the area of the vertical tail, respectively, whereas the coefficients $(A_{v(f)}/A_v)$, $A_{v(hf)}/A_{v(f)}$ and $K_{vh}$ are obtained from Roskam [p.388–p.390] [25].

### 2.5. Calculation of the Vertical Tail Plane Geometry

The calculation of the vertical tail size is performed via a MATLAB® script that loops the whole process described in Section 2.4. The needed aircraft parameters are obtained from the aircraft designs calculated in MICADO. For atmospheric parameters, the ICAO standard atmosphere [26] at sea level is used. Depending on the wiring configuration selected (on-wing or cross wiring), the yawing moments resulting from the thrust of the remaining engines and the drag from the inoperative propellers are determined. Then, starting from a simplified original geometry of the VTP, the initial values for the coefficients and derivatives are calculated. In order to be able to utilize the aforementioned diagrams from Roskam Part VI [25], they were evaluated at discrete points and implemented as tables, splines or polynomial equations and interpolated between the given values. Using Roskam's statement that the maximum rudder deflection must not be more than 25°, Equation (4) gives the maximum value for $C_{n_{\delta r}}$ and via Equation (5) the maximum $C_{y_{\delta r}}$. The resulting new surface of the vertical tail plane can be derived from Equation (6).

With the assumption that the shape of the vertical tail as well as the outer profile depth and longitudinal position of the horizontal stabilizer do not change, the new vertical tail geometry and the new position of the aerodynamic centre can be calculated. These serve as an updated starting point for the calculation of the corrected coefficients and the iteration begins again. This is performed until the difference between the newly calculated surface area and the previous one is beneath the convergence limit, set to 0.1% for this study.

To carry out this investigation, some simplifications are made. First, the geometric shape of the vertical tail plane is simplified to correlate with Roskam's assumptions. The original Beechcraft 1900D, as well as the partial turboelectric aircraft described in Section 2.2, feature a vertical tail with a rather complex geometry. With the equations from Roskam it is very difficult to accurately model this shape, as they only give the surface area, the span and the sweep angle. Therefore, the vertical tail is changed to a simple trapezoid whilst maintaining the surface area and the average sweep angle of the original

tail to minimize the effects of this simplification. The position of the horizontal tail and the profile depth of the top of the vertical tail are kept constant to minimize the influence on the longitudinal stability of the aircraft. The simplified and original geometry can be seen in Figure 4.

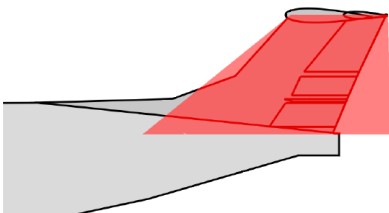

**Figure 4.** Original and simplified (red) geometry of the vertical tail.

The fuselage, on the other hand, is kept the same and held constant for all calculations. Only the size and position of the wing and the vertical and horizontal tail are allowed to change. The power-split between the different engines is also fixed, each propulsor contributes equally to the total thrust of the aircraft. Possible reductions in the power setting of the electric motors in case of engine loss are not taken into account in this study.

*2.6. Iteration Over the Whole Aircraft*

A change in the size of the vertical tail has accumulating effects on the whole aircraft—the structural mass, the CG change, and the drag of the aircraft. Therefore, a redesign of the aircraft is necessary to achieve a feasible configuration. This redesign, in turn, has an influence on the required size of the vertical tail. In order to capture these interrelated effects and to obtain a more accurate estimation of the necessary vertical tail size, the methods described in Section 2.4 are extended to take into account the changes in the overall aircraft design. This is performed in two steps. First, the MICADO environment is used to carry out a parametric aircraft design study on a range of fin volume coefficients for all four concept aircraft. Starting from a value of 0.0225, the volume coefficient is increased in steps of 0.02 up to 0.1625 and the resulting aircraft parameters, including among others the fin size, are put out into a data table. This parametric study also includes the fin volume coefficient of 0.0825 of the original Beechcraft 1900D. The data table mentioned before is used as a data source for the subsequent sizing loop of the vertical tail. Second, the iterative design loop for the sizing of the vertical tail which is executed as follows.

The initial iteration step uses the aircraft data from the database created before which belongs to the entry of the original fin volume coefficient of 0.0825. The dataset includes information on e.g., propeller positions and maximum takeoff thrust from the aircraft. Based on this, a new size for the vertical tail is calculated using the algorithm described in Section 2.5, taking into account the propeller positions and wiring options of the partial turboelectric propulsion system. The second and all subsequent iteration steps start with an interpolation of the aircraft data with respect to the previously determined size of the vertical tailplane and the aircraft data, obtained from the parameter study on different fin volume coefficients. Among others, this yields new propeller positions due to changes in wingspan and values for maximum takeoff thrust. As described for the first iteration step, this interpolated data is used to update the size of the vertical tailplane. This process is repeated until the area of the vertical tailplane does not change more than 0.1% between two consecutive iteration steps. The whole iteration process, including the steps from Section 2.5, can be seen in Figure 5.

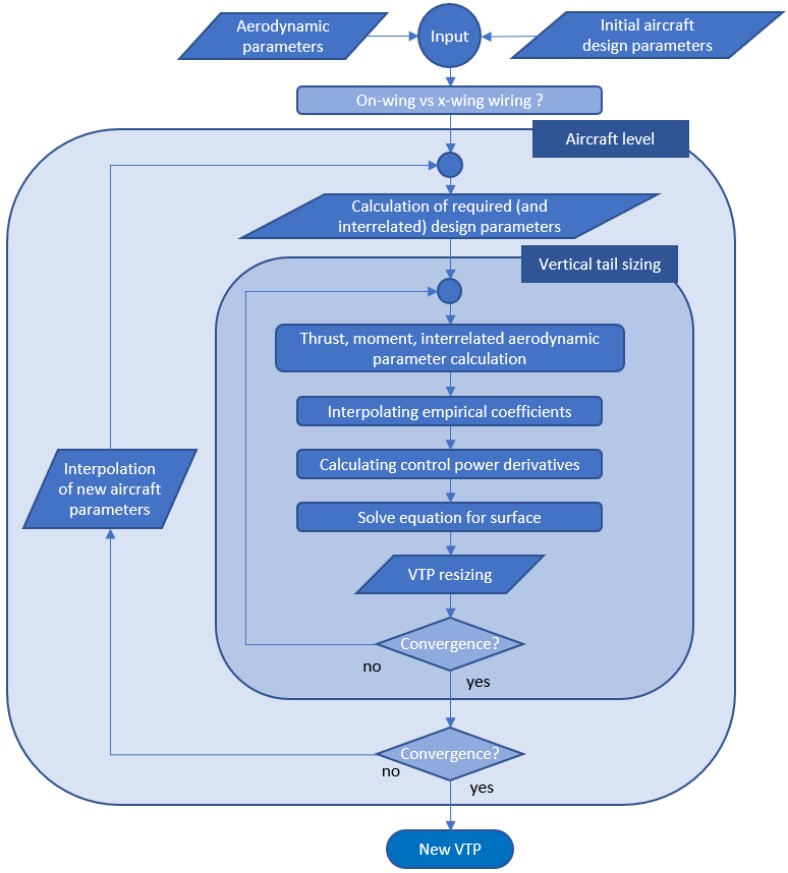

**Figure 5.** Whole aircraft iteration overview.

### *2.7. Assumptions and Restrictions*

As stated in Section 2.4, the assumptions are based on the usability of handbook methods, as validated by Ciliberti et al. [4]. Therefore, the suitability of the correction method described in this paper is only valid for configurations with similar properties. These are a high-wing configuration, a comparable tail-cone geometry and a VTP aspect ratio of around 1.0. Additionally, a VTP shape close to the simplified versions of [25] should be considered. As mentioned in the introduction, according to [6] the additional rolling motion due to the local absence of power augmented lift downstream of the propellers is neglected. As the wing tip propellers reduce the induced drag, their failure will increase the induced drag slightly. As the total reduction in induced drag through the use of a wing tip propulsion is only around 5% for the investigated aircraft [16], this effect is neglected. To reduce the complexity of this study, the propeller-induced side wash interaction on the vertical tail and the possible lift increase on the wing due to the induced velocity, are neglected.

A possible solution to reduce the VTP size for the PT2025 configurations could be the implementation of an automatic thrust control system, which is able to reduce the thrust on the electric engine quickly in case of a failure of the conventional engine, according to CS23.904 [8]. Another possibility would be the interconnection of all electric machines or the implementation of a buffer battery. This way, the electric propellers could still operate if one combustion engine fails. The downside to this would be the increased weight due to the added components and the increased complexity, which is why this case was not considered in this study.

### 3. Results

In the following section, the results of all studies are shown. First, the calculation of the required VTP sizes for the four propeller and wiring configurations is presented,

while the rest of the aircraft remains unchanged, negating the so-called snow ball effects resulting from changes to one or more aircraft components. As can be seen in Section 3.2, some configurations would require a huge VTP in order to be controllable. Therefore, only designs with realistic fin sizes are further evaluated in Section 3.3.

### 3.1. Sensitivity Study on the Vertical Tail's Volume Coefficient

In order to evaluate the influence of changes to the vertical tail's volume coefficient, the MICADO environment is used to carry out a systematic study on this design parameter. Figure 6 illustrates the resulting aircraft designs of the partial turboelectric aircraft with optimized propeller positions and a volume coefficient of 0.04253 and 0.14253. This also explains why it was not possible to calculate a converged aircraft design for a volume coefficient of 0.16253 or greater, as this leads to a large vertical tail and not physically feasible aircraft. The results of this study, in terms of the operating empty mass and the required trip fuel of the aircraft, can be seen in Figure 7.

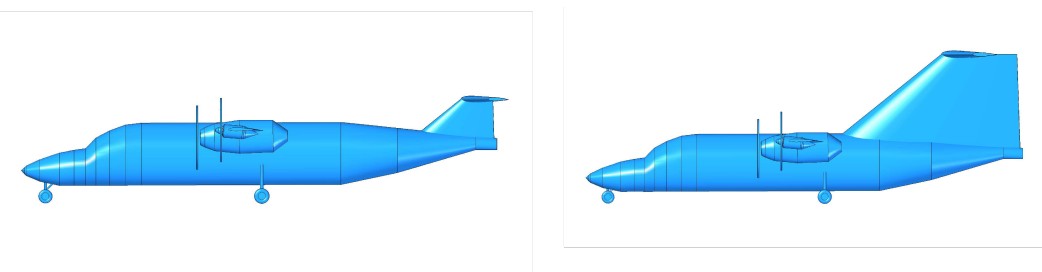

**Figure 6.** Partial turboelectric aircraft with optimized propeller positions and a volume coefficient of the vertical tail of 0.04253 (**left**) and 0.14253 (**right**).

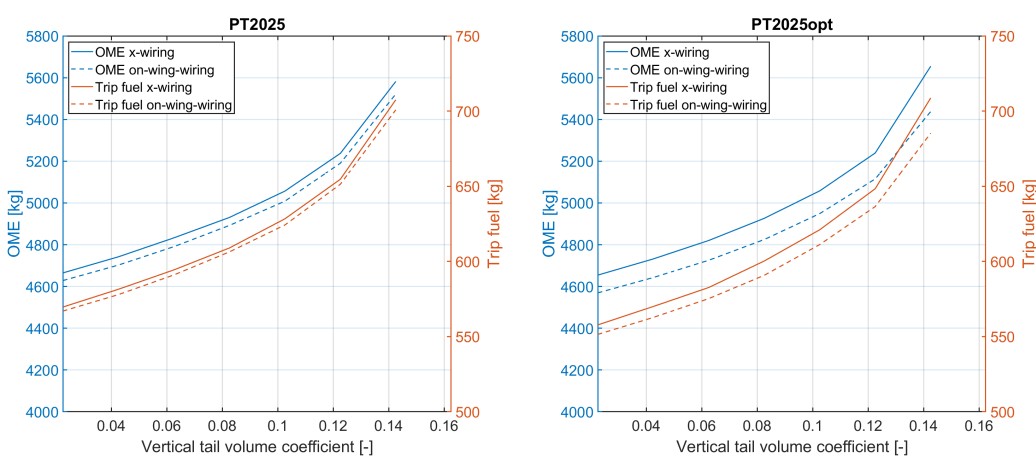

**Figure 7.** Influence of the vertical tail's volume coefficient on the OME (blue) and the trip fuel (red) for the PT2025 (left) and PT2025opt (right) configurations.

The OME and trip fuel were chosen to show the cascading effects caused by a change in the volume coefficient. A bigger volume coefficient results in a larger vertical tail, which increases the OME of the aircraft. This leads to a larger wing needed to generate the required lift, further increasing the empty mass. For a constant volume coefficient, the larger wing also means a larger vertical tail. This loop continues until an equilibrium is reached and explains the non-linearity of the OME seen in Figure 7. This effect is even more prominent for the mission fuel, as the fuel burn not only depends on the mass of the aircraft, but also on the drag. A larger vertical tail means a bigger wetted surface and an increased viscous drag.

The slight differences between the on-wing and cross-wiring options result from the different cable lengths and masses. This effect is more visible for the PT2025opt aircraft,

as the gas turbines are located further outward on the wing. This increases the distance to the electric engine on the opposite wing, while minimizing the distance to the electric engine on the same wing.

### 3.2. Size of the Adjusted Vertical Tail

The results for the resizing of the vertical tail are shown for all four configurations, in form of the surface areas and corresponding volume coefficients, are given in Table 3. The difference in the surface areas of the original vertical tails for the different wiring options results from the distinct cable lengths, as described in Section 2.3. The cross-wiring option needs longer cables than the on-wing wiring and therefore has a higher OME and MTOM. The resulting aircraft needs a larger wing to account for the increased lift demand. Since MICADO is set to calculate the size of the vertical tail based on a fixed volume coefficient this results in a larger vertical tail, as shown in Equation (3).

As can be seen in Table 3, the size of the vertical tail strongly depends on the chosen wiring option and the configuration. In case of the PT2025 aircraft, the required vertical tail area for the on-wing wiring (Configuration 1) is twice the surface of the cross-wiring option (Configuration 2), since the remaining thrust in the case of a gas turbine failure is accumulated on one side of the aircraft. The large lever arm of 8.6 m of the electric engine at the wing tip, in addition to the lever arm of 2.5 m of the turboprop engine, increases the yawing moment compared to the conventional reference for which the vertical tail was originally designed. For the cross-wiring option the residual yawing moment is still larger than for the reference aircraft, as the span-wise distance between the integration positions of the electric and the turboprop engine is 2.4 times the distance between the turboprop engine and the fuselage centreline in the reference configuration. This equalizes the fact that the electric engine only produces half the thrust of the turboprop engine in the conventional reference aircraft. As expected, the more outward position of the gas turbine in the PT2025opt aircraft results in a larger difference in the VTP size between the two wiring options. Here, the vertical tail in the on-wing wiring (Configuration 3) case is almost nine times as large as the one for the cross-wiring (Configuration 4) and even supersedes the wing, which has a surface area of 32.15 m$^2$. The cross-wiring option, on the other hand, results in a vertical tail smaller than the original one, because the distance between the electric engines at 8.3 m and the turboprop engine at 6.0 m is smaller than the lever arm in the reference aircraft, while the electric engine also produces only half the thrust.

**Table 3.** Surface area and volume coefficients of the original and resized vertical tail for the four considered aircraft and wiring configurations.

|  |  | Original VTP | Resized VTP | Change |
|---|---|---|---|---|
| **Configuration 1** | $S_v$ | 6.75 m$^2$ | 27.72 m$^2$ | +310.7% |
|  | $\bar{V}_v$ | 0.083 | 0.247 | +197.6% |
| **Configuration 2** | $S_v$ | 6.77 m$^2$ | 13.47 m$^2$ | +99.0 % |
|  | $\bar{V}_v$ | 0.082 | 0.147 | +79.3% |
| **Configuration 3** | $S_v$ | 6.68 m$^2$ | 36.38 m$^2$ | +444.6% |
|  | $\bar{V}_v$ | 0.083 | 0.288 | +347.0% |
| **Configuration 4** | $S_v$ | 6.87 m$^2$ | 4.14 m$^2$ | −39.7% |
|  | $\bar{V}_v$ | 0.083 | 0.053 | −36.1% |

### 3.3. Results of the Total Aircraft Iteration

The results of the vertical tail, as seen in Table 3, show that the on-wing wiring option is not suitable for both configurations, as it would lead to large VTPs, which significantly increase the overall weight and drag of the aircraft. The aircraft with on-wing wiring are therefore neglected in the following calculations that account for the aircraft changes and only the cross-wiring options are studied further. Additionally, MICADO could not reach convergence for aircraft with volume coefficients above 0.1425 since this leads to an aircraft that is not physically feasible, as can be seen in Figure 6.

For the PT2025 configuration, the aircraft VTP optimization was unable to reach a converged result, as the increase in mass, required thrust and change in CG, due to

the larger VTP, led to a further increase in the necessary tail area. After one iteration, the updated VTP area was 20.26 m$^2$, already outside MICADO's calculated design space. Therefore, no results for this configuration can be shown.

For the PT2025opt configuration, the aircraft level calculation could be conducted, with the results displayed in Figure 8. In contrast to the method described in Section 3.2, the position of the horizontal tail was allowed to change, since the whole aircraft design and horizontal stability were taken into account. The resulting surface area and volume coefficient, as well as the change in OME and mission fuel are given in Table 4.

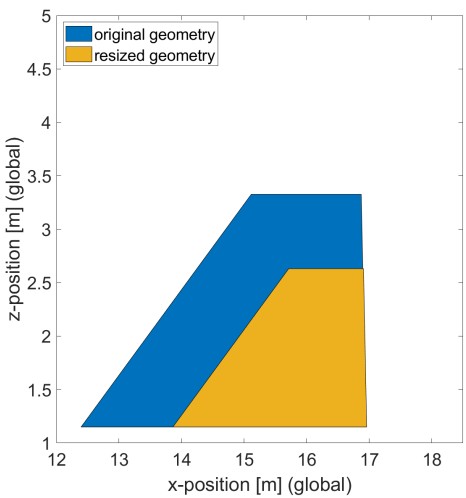

**Figure 8.** Original (blue) and resized (amber) vertical tail for the PT2025opt aircraft with cross wiring (Configuration 4).

Compared to the considerations of the isolated vertical tail, the resulting fin surface area is decreased. The larger distance between the aerodynamic centres of the wing and the vertical tail, as well as the changes in the whole aircraft explain the difference between the relative change of the fins surface area and the volume coefficient. The mass savings for the vertical tail of 79.6 kg offset the extra mass of the cables of 67.7 kg, resulting from the cross-wiring configuration. The cascading effects described in Section 3.1 also explain why the decrease in the OME is 182.0 kg, 102.4 kg more than the mass savings for the vertical tail. The additional drag reduction due to the smaller vertical tail leads to a combined fuel saving of 4.7% for the design mission of 510 NM.

**Table 4.** Comparison of VTP, OME and mission fuel of the original and optimized PT2025opt aircraft with cross wiring (Configuration 4), taking into account the iteration of the whole aircraft.

|  | Original Configuration 4 Aircraft | Configuration 4 Aircraft with Adjusted Fin | Change | |
|---|---|---|---|---|
| VTP surface | 6.87 m$^2$ | 3.18 m$^2$ | −3.69 m$^2$ | −53.7% |
| Volume coeff | 0.083 | 0.045 | −0.038 | −45.8% |
| VTP mass | 146.6 kg | 66.9 kg | −79.6 kg | −54.3% |
| OME | 4926.6 kg | 4744.6 kg | −182.0 kg | −3.7% |
| Trip fuel | 600.3 kg | 571.8 kg | −28.5 kg | −4.9% |

## 4. Conclusions

The sizing of the vertical tailplane based on volume coefficients is clearly inadequate when it comes to novel powertrain concepts. The presented method allows to amend an existing preliminary design tool with a more detailed assessment of the vertical tailplane design without the need of changing the design process itself. This is performed by resizing the vertical tail with handbook methods that account for specific concepts, such as outboard positioned propellers. Within the given restrictions the research showed that the

introduction of wing tip propellers has a significant influence on the sizing of the vertical tail. Out of the four configurations investigated in this study, only one configuration leads to a smaller vertical tail, reducing the VTP area by 53.7%, compared to the initial volume coefficient-based sizing of the PT2025opt configuration performed in MICADO, whilst fulfilling the necessity to counteract the residual torque resulting from the loss of one gas turbine. This leads to a reduction in OME of and 3.7% and a 4.9% lower fuel consumption over the given design mission. For the PT2025 configuration the electrification of the powertrain leads to unrealistic large vertical tailplanes that supersede the reference wing area.

The results of this work present a simple method to be integrated in the preliminary aircraft design and the necessity to do so. In our specific high-wing, T-tail configuration with wing tip propulsors, the possibility to reduce the VTP size and therefore the possibility to reduce fuel consumption and emissions of the investigated configuration was shown. As this method has the potential to improve the preliminary design of unconventional aircraft, further research should be conducted to test its suitability for different configurations, while possibly substituting the used handbook methods by advanced methods, depending on the required computation time.

**Author Contributions:** Conceptualization, A.A., A.B., E.S. and A.S.; methodology, A.A.; software, A.B.; data curation, P.S. and C.Z.; writing—original draft preparation, A.A., A.B, P.S. and C.Z.; writing—review and editing, A.A., A.B., P.S., C.Z., E.S. and A.S.; supervision, E.S. and A.S.; All authors have read and agreed to the published version of the manuscript.

**Funding:** This research was supported by Federal Ministry for Economic Affairs and Energy on the basis of a decision by the German Bundestag under Grant Agreement No. 20E1916A and 20E1916F.

**Institutional Review Board Statement:** Not applicable.

**Informed Consent Statement:** Not applicable.

**Data Availability Statement:** This research was conducted with data compiled by the "GNOSIS" consortium. It is yet available in various publications, this paper references. Please check the References, or contact the researchers for access to data.

**Conflicts of Interest:** The authors declare no conflict of interest. The funders had no role in the design of the study; in the collection, analyses, or interpretation of data; in the writing of the manuscript; or in the decision to publish the results.

## Abbreviations

The following abbreviations are used in this manuscript:

| | |
|---|---|
| CG | Centre of gravity |
| CS | Certification specification |
| CLT | Critical loss of thrust |
| DEP | Distributed electric propulsion |
| EASA | European Union Aviation Safety Agency |
| ICAO | International Civil Aviation Organization |
| MICADO | Multidisciplinary Integrated Conceptual Aircraft Design and Optimization |
| MTOM | Maximum take-off mass |
| NASA | National Aeronautics and Space Administration |
| OEI | One engine inoperative |
| OME | Operating mass empty |
| TLAF | Top level aircraft functions |
| TLAR | Top level aircraft requirements |
| UNICADO | University Conceptual Aircraft Design and Optimization |
| VTP | Vertical tailplane |

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
