# Peer review of "Influence of Electric Wing Tip Propulsion on the Sizing of the Vertical Stabilizer and Rudder in Preliminary Aircraft Design"

_aerospace, doi:10.3390/aerospace10050395_

Round 1

Reviewer 1 Report

This paper deals with influence of the propulsion on vertical tail design. Authors provide information about the process carried out with MICADO (software for aircraft design). Four configurations have been investigated and the most promising one has been detected. 

I would like to thank the authors for the effort. However, I suggest some modifications in order to improve the scientific soundness and the readability of the paper.  

Firstly, the title of the paper should be changed since the words distributed propulsion suggest another kind of the powertain architecture. I guess that the focal point is the wire configurations (on-wing and x-wing).

The sizing of the vertical tail is a well-known problem and it has been widely investigated in the past. In my opinion, the references reported do not represent the state of the art about the topic.

- Books of S. Gudmundsson, E. Obert, E. Torenbeek

- ESDU technical notes (i.e. NACA TN 2358, Effect of vertical-tail area and length on the yawing stability characteristics of a model having a 45 deg sweptback wing).

- Ciliberti et al., Aircraft directional stability and vertical tail design: A review of semi-empirical methods, Progress in Aerospace Sciences

About the configurations analysed, the Beechcraft 1900D was the reference but I would like to ask more information about the design exploration process about the other three configurations considered.

The wiring configurations must be explained better (section 2.2)

Title of Section 2.5 is probably incomplete ("Calculation of the vertical tail plane" size (?))

I didn't understand the sentence "The position and the profile depth of the top of vertical tail are kept constant..." (line 246-248)

In the section 2.6, it is not clear to me the first step of the iteration: 

Line 269: The initial iteration step uses aircraft data, such as propeller positions and maximum takeoff thrust from the aircraft configuration with the
original volume coefficient of 0.0825 
(Conventional Beechcraft configuration?)

Based on this data, a new size for the vertical tail is calculated, taking into account the propeller positions and wiring options of the partial turboelectric propulsion system - What is the input of the second step? 

Probably, it could be useful to add some notes to explain better the process.

Line 284-285: the calculation of the required VTP sizes for the four propeller and wiring configurations is presented, while the rest of the aircraft remains unchanged, neglected the so called snow ball effects resulting from changes to one or more aircraft components - I think it is not useful report results without the snowball effects (from Aircraft Design point of view)

About Section 3.2, the structure is not so clear in my opinion. I suggest to show all configurations investigated, with a different nomenclature (Reference, Config. 1, Config. 2, and so on)

How the directional stability is accounted for? Also for section 3.3

It will be interesting having more information about weight breakdown, in particular information about the cables weight estimation should be really useful

Table 2: what is the optimization problem statement?

Conclusion section does not reflect the content fo the paper, in my opinion. It must be related to Abstract and Introduction. 

Do not report any reference in this section (line 378).

I guess the lineas 384-386 are a good starting point for this section, while notes about MICADO should be reduced (also in section 3.3).

Reviewer 2 Report

This paper covers an interesting topic and is well written, for which I congratulate the authors. The impact of DEP on sizing of control/stability surfaces is definitely a very relevant field of research.

However, I have a hard time justifying why this should be published as a journal article. What is the key novelty/contribution of this research? The conclusions say that “the research shows that the introduction of distributed electric propulsion and wing tip propellers has a significant influence on the sizing of the vertical tail” – but this is a trivial statement. What can the reader learn from this study, that is not already addressed in existing literature? The fact that there is not a *single* reference to previous work on the effect of DEP on VTP sizing suggests that the authors have not reflected upon the added value of this study as a journal publication.

One takeaway that comes to mind as a possible “key contribution” is: cross-wiring the motors in DEP configurations may be a good solution to limit VTP size despite the required increase in cable weight. However, here I have my doubts regarding the representativeness of results, and whether the conclusions are valid in a broader sense:

·       Is the use of volume coefficients accurate for DEP configurations? If I understand correctly, volume coefficients are based on data of “conventional” aircraft configurations. But for DEP aircraft the tail may be sized by different effects (e.g. destabilizing effect of many propellers ahead of CG, distributed slipstream effects, low (or high, with tip props) yawing moments in case of failure…). The last sentence of the conclusion confirms this, since it states that volume-coefficient approach is not suited to capture the effects of this unconventional configuration. How valid are the results then?

·         Why are only two wiring options and constant thrust setting evaluated? I understand that the scope must be limited somehow, but is this a realistic simplification? Wouldn’t double redundant wiring and different thrust shares on the propellers (as suggested in the conclusions) be a more realistic case, leading to very different results?

·         Can the conclusions be drawn in a broader sense; e.g. with more propellers, or if all 4 props are placed inboard? I feel the current conclusion is very configuration specific.

For this work to be a good journal contribution I think some additional work is required, and at the very least the objectives & limitations of the results as discussed above must be clarified to the reader. If the authors do intend to submit a major revision, I have some additional comments:

- The introduction should be re-written to include relevant literature and indicate the objective/novelty of this work, as mentioned above. I suggest having a look at e.g. the following papers:

o   Biser et al. “Design Space Exploration Study and Optimization of a Distributed Turbo-Electric Propulsion System for a Regional Passenger Aircraft”, 2020

o   Hoogreef and Soikkeli. “Flight dynamics and control assessment for differential thrust aircraft in engine inoperative conditions including aero-propulsive effects”, 2022

o   Nguyen Van et al. “Towards an Aircraft with Reduced Lateral Static Stability Using Electric Differential Thrust”, 2018

o   Kou et al., “Powered Yaw Control for Distributed Electric Propulsion Aircraft: A Model Predictive Control Approach”, 2021

And maybe some papers on the impact of propeller slipstreams on the tail are also relevant:

o   Schroijen et al., “Propeller slipstream investigation using the Fokker F27 wind tunnel model with flaps deflected”, 2008

o   Zhao et al., “Propeller Rotation Effects on Pitching Moment for Transport Aircraft with Conventional Tail”, 2023

- End of abstract: “leads to a converged and sensible result”. What does “sensible” mean?

- The design tool uses a physics-based model for wing weight estimation, and “aerodynamic and structural analyses revealed a further outboard location of the gas turbines being more beneficial”. Does this model also account for aeroelastic effects and landing loads? If it only accounts for 2.5g pull-up, I imagine the optimum spanwise propeller position would be very different.

- Please specify if the volume coefficient is input to the model, or if it is calculated.

-  Sections 2.5 and 2.6 can be shortened because they mostly describe the implementation, which is user-specific and therefore not so relevant. I suggest focusing on the models used, the assumptions made, and especially the limitations of the models.

- Sec. 3.3: The optimization was not able to converge… leading to a volume coefficient “outside MICADOs calculated converged design space”. What do you mean? Why does a design not converge? Does the MTOM diverge and the aircraft is not physically possible?

- Table 2: change wording of “original aircraft” and “optimized aircraft”; not sure if you mean the B1900D or the one with inboard motors, and not sure which wiring is considered.

Reviewer 3 Report

This paper shows numerical results obtained from the study of distributed propulsion on a typical fixed-wing aircraft. The subject is relevant to the development of new design criteria for planes with distributed propulsion, regarding flight conditions with one engine out. Comments and suggestions to improve the paper are given below:   Main comments about the contents of the paper: -The abstract is too long and should be reduced (e.g. from the current 27 lines to about 15 lines).   The authors claim that: "Since aerodynamic and structural analyses revealed a further outboard location of the gas turbines being more beneficial", but the authors should provide additional information to support this statement, given its relevance to the results of the research. Typically, moving the motors away from the fuselage could create higher loads at the wing root, thus increasing the aircraft's empty weight.   The authors should point out that the analysis is applicable only to aircraft typically limited by rudder deflection during OEI conditions. For some aircraft, or even other configurations (flaps extended, landing gear down), the OEI limiting condition might be aileron authority, or even the stall speed might be achieved before the Vmc, if the aircraft is heavy.    One of the weak points in this paper lies in the fact that "Possible reductions in the power setting of the electric motors in case of engine loss are not taken into account in this study". By reducing the power in the electric motor, the propulsive asymmetry is reduced, thus decreasing Vmc. However, it was assumed that Vmc = 1.2Vs. Therefore, the power reduction required might not be large, but just enough to reduce Vmc to Vs, which is the minimum velocity achievable anyway.   Section 4 "Conclusion" should be adjusted with verbs in the past tense: "The research showed..", "The results also showed...", etc.     Other comments, mostly regarding format and English language:
  "These represent a statistic approach" Statistical   "and correlating aerodynamic drag." Correlated   "two gas turbines at it original locations " at their original locations   "are already investigated" have been   "is be given" is given   “Afterwards, adjustments to the configuration, such as moving the wing from a low to a high position and a corresponding change of the landing gear position as well as an increase of the engine performance and an application of scaling factors on the mass estimation methodologies as described in Section 2.3 were made. “ This sentence is Too long. Please use periods and commas for clarity.   “While mass estimation of most “ “Whereas” is preferred here instead of “while”   “For the resizing of the vertical stabilizer” Place a comma after this sentence   In eq 4, The denominator terms should be grouped with parentheses, for clarity.   “The the dynamic pressure” The dynamic pressure   “the yaw moments resulting from” The yawing moments   “With the assumption that the shape of the vertical tail as well as the outer profile depth and longitudinal position of the horizontal stabilizer does not change” do not change   “horizontal stabilizers position” horizontal stabilizer’s position   “Bot the simplified, as well as the original geometry can be seen in Figure 4.” The simplified and the original geometries can be seen in Figure 4.   “Only the wing position and size, as well as the empennage are allowed to change” By empennage, you mean Vertical and horizontal tail surfaces, correct? Please make that clear in the text.   “to achieve an feasible configuration” a feasible   “methods as described in the previous chapter” methods described Previous section, not chapter. Ideally, specify the number of the section.   “The results in terms of the OME” Define OME here. It is defined in the following paragraph, but the first use seems to be here.   “gas turbine failuer” failure    

Round 2

Reviewer 1 Report

Nothing to add. Thanks to the authors for the effort to improve the paper

Author Response

Dear Reviewer,

thank you for the interest in our paper, the time you invested in the review process and the resulting improvement in quality of our work.

Sincerely,

The authors.

Reviewer 2 Report

See comments attached

Reviewer 3 Report

The authors have addressed all of my concerns about the paper. I believe it can be accepted in its current form.

Round 3

Reviewer 2 Report

Thank you for addressing my previous comments.